# The Japanese version of the Fear of COVID-19 scale: Reliability, validity, and relation to coping behavior

**Koubun Wakashima**[1]*, **Keigo Asai**[2], **Daisuke Kobayashi**[1], **Kohei Koiwa**[1], **Saeko Kamoshida**[1], **Mayumi Sakuraba**[1]

1 Graduate School of Education, Tohoku University, Sendai, Japan, 2 Graduate School of Education, Hokkaido University of Education, Hokkaido, Japan

* kobun.wakashima.d3@tohoku.ac.jp

**Data Availability Statement:** All relevant data are within the paper and its Supporting Information files.

## Abstract

COVID-19 is spreading worldwide, causing various social problems. The aim of the present study was to verify the reliability and validity of the Japanese version of the Fear of COVID-19 Scale (FCV-19S) and to ascertain FCV-19S effects on assessment of Japanese people's coping behavior. After back-translation of the scale, 450 Japanese participants were recruited from a crowdsourcing platform. These participants responded to the Japanese FCV-19S, the Japanese versions of the Hospital Anxiety and Depression scale (HADS) and the Japanese versions of the Perceived Vulnerability to Disease (PVD), which assesses coping behaviors such as stockpiling and health monitoring, reasons for coping behaviors, and socio-demographic variables. Results indicated the factor structure of the Japanese FCV-19S as including seven items and one factor that were equivalent to those of the original FCV-19S. The scale showed adequate internal reliability (α = .87; ω = .92) and concurrent validity, as indicated by significantly positive correlations with the Hospital Anxiety and Depression Scale (HADS; anxiety, r = .56; depression, r = .29) and Perceived Vulnerability to Disease (PVD; perceived infectability, r = .32; germ aversion, r = .29). Additionally, the FCV-19S not only directly increased all coping behaviors (β = .21 - .36); it also indirectly increased stockpiling through conformity reason (indirect effect, β = .04; total effect, β = .31). These results suggest that the Japanese FCV-19S psychometric scale has equal reliability and validity to those of the original FCV-19S. These findings will contribute further to the investigation of various difficulties arising from fear about COVID-19 in Japan.

## Introduction

As of September 2020, infectious disease caused by the novel coronavirus (coronavirus disease 2019, or COVID-19) continues to spread on a global scale. This most recently discovered coronavirus emerged in the city of Wuhan in Hubei Province of China in December 2019 [1]. As of September 9, 2020, approximately 27 million people (confirmed cases) have been infected worldwide; the number of deaths has risen to approximately 900,000 [2].

**Funding:** The authors received no specific funding for this work.

**Competing interests:** The authors have declared that no competing interests exist.

The first case of infection in Japan was confirmed on January 16, 2020: a Chinese man residing in Kanagawa Prefecture who had lived for some time in Wuhan. Subsequently, the spread of infection among local residents who had never visited Wuhan and cluster infections occurred in Japan, which led to a rapid rise in the number of infected people, particularly in Tokyo and Osaka. On April 16, 2020, the Japanese government declared a national state of emergency [3]. By September 9, 2020, the number of infected persons nationwide had risen to 72,726, leading to 1,393 deaths. Such a widespread social effect created confusion among the general public [4]. More specifically, reactions such as bulk purchases (stockpiling) of face masks and sanitizers soared, creating secondary disruptions [5].

Because fear of the COVID-19 epidemic can adversely affect disease management [6], fear of COVID-19 should be assessed appropriately. Some researchers have already conducted studies of people's fear of infection with COVID-19 that is currently spreading worldwide [7–9]. The Fear of COVID-19 Scale (FCV-19S) was developed with the aim of quelling fear of COVID-19 and for other goals [10, 11]. The FCV-19S is positively associated with anxiety and depression, and with perceptions of vulnerability to infection.

The FCV-19S, based on the Protection Motivation Theory [12], has a unidimensional factor structure [10, 13]. The FCV-19S has been confirmed to have reliability and validity in various countries such as Bangladesh [14], Iran [10], Israel [15], Italy [16], New Zealand [17], Russia and Belarus [18], Saudi Arabia [19], Turkey [20], and Vietnam [21]. It has come to have more widespread use than other corona-related measures [13].

Results obtained using FCV-19S have been found to be associated with various factors including socio-demographic and residential environments. Being female, older, smoking, using health care services for COVID-19-related stress, and worries related to lockdown are factors associated with higher FCV-19S [14, 18, 21–23].

In addition, fear and anxiety can affect social behavior. Among the psychological responses and coping behaviors of people following the 2009 influenza A (H1N1) outbreak, those with less resistance to uncertainty were more likely to use coping behaviors aimed at releasing their emotions [24]. In the current context, fear of COVID-19 is not only positively associated with prevention behaviors [25, 26] and health-related behaviors such as increasing alcohol and tobacco use [21]; it is also associated with bulk buying behaviors [27].

However, these studies have not examined reasons underlying those behaviors. In Japan, where peer pressure is high, some people wear masks not only because it is necessary as a preventive behavior, but also because they are worried about how others will think of them if they do not wear a mask [28]. Therefore, reasons of two types are assumed for actions: self-determining reasons, by which a person decides to act independently out of a sense of need, and conformity reasons, by which a person attunes self-behavior to the actions of surrounding people out of fear or anxiety. This study therefore examines a model of the relation between fear of COVID-19 and coping behavior with reasons of two types mediating fear and coping behavior.

The purposes of this study were twofold. First, this study translates FCV-19S, established by Ahorsu et al. [10], into Japanese and assesses its reliability and validity in Japan based on a procedure equivalent to that used by Ahorsu et al. [10]. Secondly, this study develops a model of COVID-19 fear effects on coping behaviors, including reasons for the behaviors.

## Materials and methods

### Procedure

Data were collected in Japan. The participants were 450 residents of Japan (291 men and 159 women) aged 18 and older with average age of 48.13 years (SD = 14.04). They were recruited

through an online service provided by Yahoo (https://crowdsourcing.yahoo.co.jp/), a major crowdsourcing service in Japan managed by Yahoo Japan Corp. The online service coordinates requests from clients with crowdsourcing workers. The current research was posted by the authors as a psychology research project in the "questionnaire" category on the Yahoo information board. The number of participants in the survey was set to be terminated when 450 participants had been recruited. The number of participants was determined based on the number of participants in another FCV-19S study [10] to exceed the minimum sample size required by CFA and SEM [29, 30]. The participants read the study descriptions and agreed to take part in it by opting into the study themselves. Participants who completed the survey were assigned 105 points (about one dollar) to be used at a specific store. The survey questionnaire was administered on April 18 and 19, 2020.

## Survey description

Socio-demographic variable: The survey questionnaire asked for free answers from each respondent for sex, age, nationality, and residential area (city and prefecture). This item was followed by a multiple-choice question asking about the participants' health condition at the time, which allowed for many responses such as 1 = "in normal condition," 2 = "having a fever of 37.5°C or higher," 3 = "having a sore throat," 4 = "having deep fatigue," 5 = "having a cough," and 6 = "having other symptoms" [31]. The subsequent analysis combined answers that included at least one from choices 3 through 6 into one group, with three other groups including 1 = "in normal condition," 2 = "having a fever of 37.5°C or higher," and 3 = "having other symptoms." The next question categorized the diseases being treated at the time into respiratory diseases, mental disorders (anxiety disorder, depression, and other mental disorders), and other diseases and asked the participants to indicate whether they had a disease, and if they did, to give the specific name(s) of the disease(s). The analysis combined the types of diseases into one group and divided the answers based on whether or not the participants had a disease being treated at the time. Subsequent questions asked all participants whether they smoked and asked female participants whether they were pregnant. All participants in this survey were Japanese citizens. For this study, the variables of sex, age, health condition at the time, diseases being treated, and smoking habits, which were expected to be associated with FCV in earlier studies [15, 18, 20–22], were used for the analysis.

Working status: The questionnaire asked whether the participants worked at a job at the time. If they did, the questionnaire asked them to describe their job. Participants were also asked whether they were able to work from home, how they commuted, and at what hierarchical level in the company their job was positioned. After combining responses indicating whether the participants worked at a job and whether they were able to work from home, three categories were created for analyses, including a group being able to work from home, one unable to work from home, and a group of participants not working at a job.

Family composition: Participants were asked about the number of family members living with them, their relation to the participant, their age, whether they had respiratory or other diseases, their smoking history, and their pregnancy status (only female family members residing with the participant). In addition, participants were asked about changes in the amount of conversations and conflict of opinions in the family living together in the last month. Only people living with family members who were expected to be associated with FCV in earlier studies [32, 33] were included in the analysis.

Sources of information about COVID-19: Participants were asked a multiple-choice question about media that they regarded as a valuable source of information about COVID-19. They were also asked to rank such information sources from first to the third based on their

importance. Specific sources of information indicated in the answers were the following: 1 = newspaper; 2, news on TV; 3, talk shows on television; 4, websites of public organizations; 5, news on the internet; 6, Twitter; 7, Facebook; 8, Instagram; 9, other social networking services ("SNS"); and 10, other media. The analysis used only the most prioritized information sources and combined choices 6–9 into a single group called "SNS."

Presence of persons infected with COVID-19 around the participant: Participants were asked whether anyone with whom they were acquainted had contracted COVID-19. Any respondent acquainted with an infected person was asked to describe their relation. To assess the status of COVID-19 infection for individuals around the participant, the survey asked about an infected person's residence: 1, "in the same prefecture"; 2, "in the same municipality"; or 3, "in the same district" as the participant; or 4, no one was infected around the participant. The analysis combined responses for 1 through 3 into one group, designated as 1, "there is an infected person nearby" and the rest 2 "there is no infected person nearby."

Measuring fear of COVID-19: The study used a Japanese translation of FCV-19S developed by Ahorsu et al. [8]. The following describes the procedure for translating the scale. First, permission to produce a Japanese translation of FCV-19S was obtained from Dr. Amir H. Pakpour, one author of the original article on FCV-19S. A translation agency performed the translation. The author of this report and the translation agency then modified Japanese expressions used for the items of the scale. A Korean psychologist fluent in Korean, English, and Japanese subsequently performed a reverse translation of the Japanese translation without seeing the original version. Finally, a native speaker of English compared this reverse translation and the original version of the scale and confirmed that they were fundamentally the same. The Japanese version was therefore finalized. The Japanese version of the scale that was completed consisted of seven items in the same manner as the original. These seven items were made a provisional Japanese version of the Fear of COVID-19 Scale ("FCV-19S-J"). FCV-19S-J asks questions to be answered on a scale of 1, "I am not afraid of COVID-19 at all" to 5, "I am most afraid of COVID-19." A higher score reflects a greater fear of COVID-19 (S1 File).

Measuring anxiety and depression: The Japanese version of the Hospital Anxiety and Depression Scale (HADS) prepared by Hatta et al. [34] was used. Although the HADS [35] was designed originally for patients in nonpsychiatric hospital clinics, it has been found to be reliable and valid in general samples [34]. HADS was positively correlated with FCV-19S [10]. It comprises a total of 14 items, including seven that measure recent anxiety and another seven that measure recent depression. It seeks responses from four choices each. Higher scores denote severer anxiety or depression experienced recently. Both the measures of anxiety and depression are expected to show positive correlation with FCV-19S-J.

Measuring perception of vulnerability to infection: The study used the Japanese version of the perceived vulnerability to disease (PVD) scale developed by Fukukawa, Oda, Usami, and Kawahito [36]. The PVD scale [37], which has been demonstrated as positively correlated with FCV-19S [10], consists of 15 items comprising 7 items of susceptibility to infection and 8 items of germ aversion. The response to each item is selected from seven choices from 1 = "strongly disagree" to 7 = "strongly agree." A high score represents high susceptibility to infection or high germ aversion. Both the measures of susceptibility to infection and germ aversion are expected to represent positive correlation with FCV-19S-J.

Behavior to cope with COVID-19: Participants were asked about their actions taken to cope with COVID-19. Items for coping behaviors were produced based on a report issued by the Ministry of Health, Labour and Welfare: "What we are doing to prevent new corona infection" of the "1st National Survey for Countermeasures against COVID-19" [38]. In addition, based on reports on social issues [39], items related to social behavior were added. Each of 19 items

(e.g., "avoided places with large crowds") was rated on a six-point Likert scale ranging from 1 (not at all) to 6 (very much) (S2 File).

Reason for behavior: We asked about reasons for the behaviors above to cope with COVID-19. We developed three items as proactive reasons (e.g., "I did it because I felt it was necessary for myself") and four items as passive reasons (e.g., "I did it because other people told me to"). Each of 7 items was rated on a six-point scale ranging from 1 (not applicable at all) to 6 (highly applicable) (S2 File).

### Ethical consideration

After the first section of the survey presented a statement of the survey purpose, the form stated that participation was voluntary and that the survey was anonymous: personal information would not be disclosed to third parties. Only those who agreed to cooperate in the survey would be able to proceed to the questionnaire. Additionally, the Tohoku University Graduate School of Education's ethics committee granted ethical approval for this study (ID: 20-1-003).

### Data analysis

Statistical operations were conducted using software (Mplus 8.1 [40]) and a computer program (R 3.6.3 [41]). Analyses examined the reliability and validity of the Japanese version of the FCV-19S and the effect of fear of COVID-19 on coping behavior.

To investigate the reliability and validity of the Japanese version of FCV-19S, confirmatory factor analysis, correlation analysis, and calculation of reliability coefficients were conducted. We also conducted t-tests and analysis of variance with the socio-demographic variable as the independent variable and FCV-19S as the dependent variable.

For confirmatory factor analysis (CFA), a robust maximum likelihood estimator (MLR) was applied in this study. To test goodness of fit, we conducted the following analyses: comparative fit index (CFI), root mean square error of approximation (RMSEA), standardized root mean square residual (SRMR), and Bayesian information criterion (BIC). The cut-off values for acceptable model fit used for this study were: RMSEA $< .10$ for acceptable fit and $< .06$ for good fit; CFI $> .90$ for acceptable fit and $> .95$ for good fit; and SRMR $< .10$ for acceptable fit and $< .08$ for good fit [29, 42]. Error correlation was assumed to be related to the modified index (MI) if the goodness of fit showed an inadequate value. An MI indicates the expected parameter change if a particular specification were included in the model. Reliability was calculated for Cronbach's alpha coefficients ($\alpha$) and McDonald's omega coefficients ($\omega$). Correlations between the FCV-19S-J and other measures were established by calculating Pearson's correlation coefficients. A t-test was performed when examining the difference in means between the two groups. Analysis of variance was conducted when examining the difference in means among the three or more groups. Reported effect sizes are interpreted using Cohen's d and $\eta^2$, respectively including 95% confidence intervals.

To elucidate the effect of FCV-19S on coping behavior, an exploratory factor analysis of coping behavior and reasons for behavior was conducted. A structural equation model was tested.

In exploratory factor analyses (EFA), MLR and geomin rotation was applied for coping behavior and reason for behavior. We removed items with factor loadings lower than .35. In the structural equation model (SEM), a robust maximum likelihood estimator (MLR) was applied in this study. The same indices were used for testing goodness of fit as for confirmatory factor analysis. We assumed a direct path from FCV-19S to coping behavior and a path from FCV-19S to coping behavior by mediating the reason for behavior. Therefore, we examine not only the direct but also indirect effects of FCV-19S on coping behavior.

All statistical analyses used two-tailed tests. For all statistical evaluations, p values less than .05 were inferred as significant. Missing values were visible only for age. Therefore, pairwise deletion was used for missing data.

## Results

### Factor structure, reliability and validity of FCV-19S-J

Table 1 presents participants' basic characteristics. The present study participants were 450 participants, mostly men (65%).

Confirmatory factor analyses, as described by Ahorsu et al. [10], were used to examine the goodness of fit. Results show that the Japanese FCV-19S did not fit well (Table 2, Model 1). To improve the model fit, MI were used. The MI between items 1 and 4 (MI = 47.72) and between items 2 and 5 (MI = 35.30) were higher values. Therefore, a within factor error-covariance between items 1 and 4 (Model 2) and between items 2 and 5 (Model 3) was included. The model was modified. The model that includes error correlations of items 1 and 4 / items 2 and 5 is Model 4. Results indicate that the modified model (Model 4) was more acceptable (CFI = .943, RMSEA = .105, SRMR = .052) than Model 2 or 3. This was the final model.

The descriptive statistics of the Japanese FCV-19S and other scales are presented in Table 3. Internal consistency reliability was acceptable for FCV-19S scores ($\alpha$ = .87/ $\omega$ = .92). Correlations between the Japanese FCV-19S and the HADS and PVD were significant, ranging from r = .29 to r = .56, p < .01.

### Differences based on relevant variables

Means and standard deviations of all compared groups are presented in Table 1. No significant difference was found among the variables other than the important source of information. Groups of the important source of information differed significantly with regard to the FCV-19S-J, $F_{(6, 443)}$ = 3.469, p < .01, $\eta^2$ = .044, 95% CI [.006; .076]. Post-hoc tests revealed that the News on TV group scored higher on the FCV-19S-J than the Other groups, p = .04, d = .97, 95% CI [.38; 1.56].

### Effects of fear of COVID-19 on coping behavior

Factor analysis was conducted of coping behaviors and the reasons for these behaviors. Results show that 13 items were extracted from three factors for coping behavior (Table 4); six items from two factors were extracted for reasons for the behavior (Table 5). The descriptive statistics of these scales are presented in Table 3. Structural equation modeling (SEM) was used to investigate the effect of fear of COVID-19 on coping behavior (Fig 1). The model had acceptable fit to the data, $X^2(42)$ = 179.934, p < .05, CFI = .905, RMSEA = .085, (90% CI .073 –.098), SRMR = .057. The path from FCV-19S-J to each coping behavior and conformity reason had a small and moderate effect ($\beta$ = .206 –.358, p < .001). The path from conformity reason to stockpiling and from self-determining reason to daily attention had a small and moderate effect ($\beta$ = .113 –.119, p < .05).

An indirect effect was examined to ascertain whether the conformity reason significantly mediated the relation between FCV-19S-J and stockpiling. Results show that FCV-19S-J had a significant indirect effect on stockpiling, through the effect of conformity reason ($\beta$ = .035, p < .05) for a modest total effect ($\beta$ = .311, $p$ < .01).

**Table 1. Participants' basic characteristics.**

| Variable | N = 450 | % | FCV-19 | | t /F |
|---|---|---|---|---|---|
| | | | Mean (SD) | | |
| Sex | | | | | |
| Men | 291 | (65%) | 21.01 | 5.28 | t(311.46) = 1.24 n.s. |
| Women | 159 | (35%) | 21.68 | 5.55 | |
| Age (N = 448) | | | | | |
| ≤20s | 35 | (7.7%) | 20.77 | (5.73) | F(5, 442) = 0.46 n.s. |
| 30s | 116 | (25.8%) | 21.09 | (5.86) | |
| 40s | 104 | (23.1%) | 21.20 | (5.57) | |
| 50s | 49 | (10.9%) | 20.73 | (5.43) | |
| 60s | 117 | (26.0%) | 21.84 | (4.59) | |
| ≥70s | 27 | (6.0%) | 20.93 | (5.40) | |
| Health condition | | | | | |
| In normal condition | 415 | (92.2%) | 21.23 | 5.29 | F(2, 447) = 0.52 n.s. |
| Having a fever of 37.5˚C or higher | 30 | (6.7%) | 21.80 | 6.29 | |
| Having other symptom | 5 | (1.1%) | 19.20 | 7.89 | |
| Smoking habit | | | | | |
| Smoker | 165 | (36.7%) | 21.76 | 5.2 | t(356.59) = 1.57 n.s. |
| Non-smoker | 285 | (63.3%) | 20.95 | 5.47 | |
| Diseases being treated | | | | | |
| Disease undergoing treatment | 111 | (24.7%) | 21.68 | 6.08 | t(164.31) = 0.88 n.s. |
| No disease undergoing treatment | 339 | (75.3%) | 21.11 | 5.13 | |
| Work status | | | | | |
| Able to work from home | 109 | (24.2%) | 21.45 | 5.43 | F(2, 447) = 0.47 n.s. |
| Unable to work from home | 206 | (45.8%) | 21.39 | 5.33 | |
| Not working | 135 | (30.0%) | 20.87 | 5.43 | |
| Living with family | | | | | |
| Living with family | 356 | (79.1%) | 21.44 | 5.33 | t(141.66) = 1.40 n.s. |
| Not living with family | 94 | (20.9%) | 20.54 | 5.54 | |
| Most important source of information | | | | | |
| Newspaper | 38 | (8.4%) | 19.47 | 5.00 | F(6, 443) = 3.47** |
| News on TV | 200 | (44.4%) | 22.08 | 5.01 | |
| Talk shows on television | 20 | (4.4%) | 22.70 | 6.20 | |
| Websites of public organizations | 50 | (11.1%) | 19.96 | 4.20 | |
| News on the internet | 115 | (25.6%) | 21.17 | 5.86 | |
| SNS | 15 | (3.3%) | 21.00 | 6.15 | |
| Other (e.g. radio, Youtube, No important source of information) | 12 | (2.7%) | 17.17 | 6.49 | |
| Presence of persons infected | | | | | |
| Someone I know has the corona virus. | 8 | (1.8%) | 21.75 | 10.38 | t(7.07) = 0.14 n.s. |
| No one I know has the corona virus. | 442 | (98.2%) | 21.24 | 5.27 | |

Notes: ** p< .01; sex, smoking habit, diseases being treated, living with family and presence of persons infected were subjected to *t*-tests. Analysis of variance was applied for age, health condition, work status, and sources of information.

## Discussion

The first purpose of this study was development of the Japanese version of FCV-19S. As described by Ahorsu et al. [10], the results of factor analysis indicated a single factor structure. The α coefficient and omega coefficient in FCV-19S-J returned sufficient values. HADS

**Table 2. Factor loadings for the FCV-19S-J.**

| Items | | Standardized loadings | | | |
|---|---|---|---|---|---|
| | | Model 1 | Model 2 | Model 3 | Model 4 |
| FCV-19S-J (α = .87 / ω = .92) | | | | | |
| 1 | I am most afraid of coronavirus-19. | .590 | .556 | .562 | .526 |
| 2 | It makes me uncomfortable to think about coronavirus-19. | .744 | .728 | .688 | .670 |
| 3 | My hands become clammy when I think about coronavirus-19. | .737 | .747 | .759 | .764 |
| 4 | I am afraid of losing my life because of coronavirus-19. | .510 | .471 | .489 | .449 |
| 5 | When watching news and stories about coronavirus-19 on social media, I become nervous or anxious. | .743 | .734 | .691 | .680 |
| 6 | I cannot sleep because I'm worrying about getting coronavirus-19. | .740 | .754 | .770 | .780 |
| 7 | My heart races or palpitates when I think about getting coronavirus-19. | .819 | .835 | .843 | .857 |
| | $X^2$ | 159.994*** | 113.398*** | 126.178*** | 71.068*** |
| | df | 14 | 13 | 13 | 12 |
| | CFI | 0.859 | 0.903 | 0.891 | 0.943 |
| | RMSEA | 0.152 | 0.131 | 0.139 | 0.105 |
| | 90 Percent C.I. | 0.132–0.174 | 0.109–0.154 | 0.118–0.162 | 0.082–0.129 |
| | SRMR | 0.065 | 0.051 | 0.065 | 0.052 |
| | BIC | 7921.633 | 7864.521 | 7882.834 | 7817.045 |

Notes: ***p < .001; CFI, comparative fit index; RMSEA, root mean square error of approximation; SRMR, standardized root mean square residual; and BIC, Bayesian information criterion; α, Cronbach's alpha; ω, McDonald's omega.

indicated a significant correlation between "depression" and "anxiety"; PVD revealed significant correlation between "perceived infectability" and "germ aversion," which suggests adequate reliability and validity. In FCV-19S-J, the goodness of fit was made an acceptable value by assuming an error correlation between item 1 ("I am most afraid of coronavirus-19.") and

**Table 3. Correlations between the Japanese FCV-19S-J and other scales.**

| | | Mean (SD) | Min–max | Skewness | Kurtosis | FCV-19S-J |
|---|---|---|---|---|---|---|
| **FCV-19S-J** | | 21.25 (5.38) | 7–35 | .01 | .06 | - |
| **HADS** | | | | | | |
| anxiety (α = .87/ ω = .90) | | 6.83 (4.47) | 0–21 | .77 | .35 | .56*** |
| Depression (α = .68/ ω = .76) | | 8.81 (3.84) | 0–20 | .31 | -.11 | .29*** |
| **PVD** | | | | | | |
| perceived infectability (α = .84/ ω = .91) | | 27.06 (7.31) | 7–49 | .05 | .40 | .32*** |
| germ aversion (α = .79/ ω = .85) | | 39.16 (8.26) | 14–56 | .01 | -.25 | .29*** |
| **Coping behavior** | | | | | | |
| careful in daily life (α = .81/ ω = .85) | | 39.56 (5.95) | 13–48 | -.96 | .99 | .28*** |
| Stockpiling (α = .93/ ω = .93) | | 6.34 (2.88) | 2–12 | .09 | -.78 | .30*** |
| health monitoring (α = .64/ ω = .67) | | 11.11 (3.37) | 3–18 | -.39 | -.07 | .39*** |
| **Reason for behavior** | | | | | | |
| Self-determining reason (α = .50/ ω = .50) | | 8.68 (1.66) | 2–12 | -.45 | 1.27 | .11 |
| conformity reason (α = .77/ ω = .80) | | 11.77 (3.69) | 4–24 | -.11 | -.29 | .21*** |

Notes: ***p < .001; α, Cronbach's alpha; ω, McDonald's omega

**Table 4. Factor loadings for the coping behavior.**

| Items | | Factor loadings | | |
|---|---|---|---|---|
| | | F1 | F2 | F3 |
| **F1 Careful in daily life** | | | | |
| 2 | Avoided places with large crowds | **.855** | -.057 | -.007 |
| 3 | Avoided having a conversation or utterance in proximity to another person | **.763** | -.001 | -.075 |
| 1 | Avoided places with poor ventilation | **.739** | -.003 | .058 |
| 4 | Washed hands, gargled, or sanitized hands and fingers using alcohol | **.603** | .020 | .118 |
| 16 | Refrained from eating out | **.439** | .098 | .080 |
| 5 | Covered the mouth with a mask or handkerchief when coughing or sneezing | **.424** | .016 | .273 |
| 18 | Intentionally blocked the inflow of information | **.380** | .000 | .157 |
| 11 | Received a test to determine whether you are infected with COVID-19 | **-.340** | .205 | .048 |
| **F2 Stockpiling** | | | | |
| 8 | Purchased food in larger quantities than usual | -.015 | **.956** | .010 |
| 7 | Purchased commodities in larger quantities than usual | .027 | **.903** | -.023 |
| **F3 Health monitoring** | | | | |
| 10 | Monitored heath condition more carefully than before | .262 | -.012 | **.658** |
| 9 | Observed changes in your health condition by measuring body temperature, etc. | -.002 | .155 | **.657** |
| 13 | Did something to take your mind off of COVID-19 | .002 | .106 | **.362** |
| | F1 | - | | |
| | F2 | .336 | - | |
| | F3 | .384 | .263 | - |

item 4 ("I am afraid of losing my life because of coronavirus-19.") and between item 2 ("It makes me uncomfortable to think about coronavirus-19.") and item 5 ("When watching news and stories about coronavirus-19 on social media, I become nervous or anxious."). For the error correlations, the Saudi Arabian version [19] was on items 1 and 2, items 3 and 6, items 3 and 7, and items 6 and 7, whereas the Turkish version [20] was on items 3 and 6, items 3 and 7, and items 6 and 7. We found no study with reported error correlations between items 1 and 4, items 2 and 5 as in the results of this study. FCV-19S has a unidimensional structure. However items 3, 6, and 7 are regarded as somatic responses to COVID-19 fear; items 1, 2, 4, and 5 are regarded as representing the general level of fear [19, 43]. Therefore, the error correlations in this study are error correlations between items that indicate a general level of fear. In Japan, a state of emergency declaration was expanded to include all of Japan on April 16 [3]. This

**Table 5. Factor loadings for the reason for behavior.**

| Items | | Factor loadings | |
|---|---|---|---|
| | | F1 | F2 |
| **F1 Self-determining reason** | | | |
| 1 | I did it because I felt it was necessary for myself. | **.736** | .005 |
| 3 | I did it based on my own decision. | **.451** | -.259 |
| **F2 Conformity reason** | | | |
| 5 | I did it because other people told me to. | .004 | **.829** |
| 6 | I did it out of a fear of criticism that would be raised by other people. | -.020 | **.785** |
| 4 | I followed other people. | .057 | **.722** |
| 2 | I did it even though I did not actually think it was necessary. | -.193 | **.390** |
| | F1 | - | |
| | F2 | .02 | - |

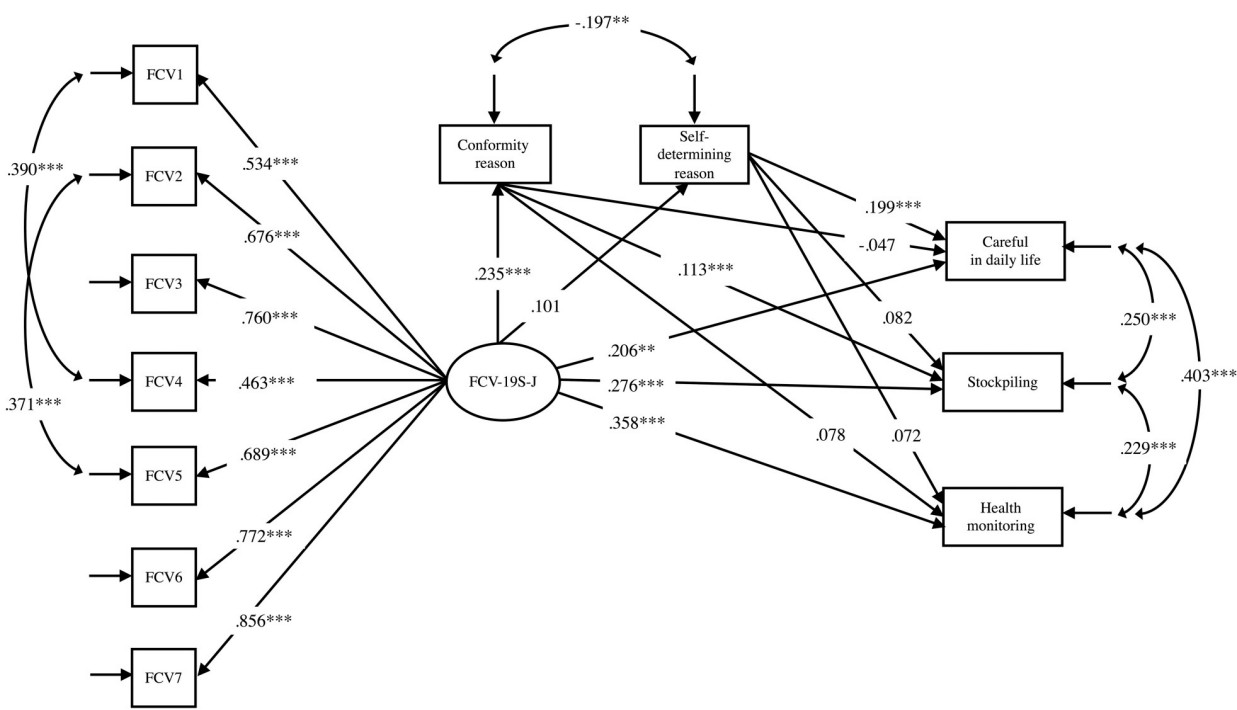

**Fig 1. Effect of fear of COVID-19 on coping behavior model.** Notes: **p< .01; ***p< .001.

event was a couple of days before the participants in this study participated in the study. This social context might have influenced the association between items of general fear rather than somatic responses.

Results of t-tests and analysis of variance suggested that participants who prioritized news on television as an information source tended to have greater anxiety related to COVID-19 than others who prioritized other information sources. The FCV-19S scores, however, were not found to be significantly different based on other factors such as the sex, age, living with family, and presence of persons infected. Earlier studies indicated that risk factors which increase the fear of COVID-19 include being female, older, smoking, using health care services for COVID-19-related stress, worries related to lockdown, and not living with a family member [14, 18, 21–23, 32]. Differences between the results of this study and earlier studies are likely to be attributable to social conditions. Because a state of emergency was declared in Japan, it is likely that everyone living in Japan is almost equally fearful, irrespective of the demographic characteristics of the study participants. Therefore, we believe that these study participants did not differ significantly in terms of their fear of COVID-19 depending on their attributes.

The second purpose of this study was development of a model of COVID-19 fear effects on coping behaviors, including reasons for the behaviors. Results revealed that fear of COVID-19 encouraged measures taken to prevent infection such as care in daily life and health conditions. This result is consistent with results indicating that FCV-19S predicted positive behavioral change (e.g. hand washing, changed travel) [26] and that it was positively associated with adherence to New Zealand's lockdown rules (e.g. maintaining the two-meter rule when out in public) [17]. Particularly, Harper [26] examined behavioral change. Therefore, various behaviors such as hand-washing and stockpiling were combined into a single variable. However, this study not only categorized coping behaviors as being careful in daily life, stockpiling, and

health monitoring; it also asked about reasons for the behaviors. Results of this study demonstrated that fear of COVID is associated with preventive behavior, and also that it is related directly to nuisance behaviors such as stockpiling. Furthermore, we have demonstrated that fear of COVID is associated with stockpiling via conformity.

The fear of COVID-19 did not affect self-determination of reasons. However, self-determination of reasons contributed to an increase in care in daily life. At least in Japan, the self-determination of reasons and fear of COVID-19 might be useful for encouraging caution in daily life.

## Limitations

This study has some limitations. Study participants were solicited using the internet. The study was able to collect people in wide-ranging age groups, but those who were willing to participate in the survey were likely to have been mentally stable, with sufficient psychological capacity to contemplate COVID-19 effects. The survey must be expanded to include participants such as medical professionals and people who have been adversely affected financially through events such as a loss of work because of COVID-19.

Moreover, the analysis used for this study used cross-sectional data, which were inadequate to verify causal relations between anxiety about COVID-19 and coping behavior. Particularly, the cross-sectional survey does not enable us to ascertain whether anxiety and fear arouses preventive behavior or whether it is aroused by performance of preventive behavior. Subsequent studies must identify relations between coping behavior and a fear of COVID-19 using a longitudinal study.

## Conclusions

Despite the limitations described above, this study has explained the factor structure of FCV-19S-J. This report is the first in Japan to describe a study identifying the relations between fear of COVID-19 and coping behavior. The environment surrounding COVID-19 changes day by day. Appropriately measuring people's anxiety and fear of COVID-19 likely to contribute to an understanding of increasing anxiety experienced by many people and to prevention of difficulties associated with COVID-19.

## Supporting information

**S1 File. The Japanese version of FCV-19S.**
(DOCX)

**S2 File. The coping behavior and reason for behavior.**
(DOCX)

**S1 Data. Anonymized data set.**
(XLSX)

## Acknowledgments

The authors would like to thank the members of our study team and the participants who took part in our study.

## Author Contributions

**Conceptualization:** Koubun Wakashima.

**Data curation:** Daisuke Kobayashi.

**Formal analysis:** Keigo Asai.

**Investigation:** Kohei Koiwa.

**Methodology:** Keigo Asai.

**Project administration:** Koubun Wakashima.

**Resources:** Daisuke Kobayashi, Kohei Koiwa, Saeko Kamoshida, Mayumi Sakuraba.

**Software:** Keigo Asai.

**Supervision:** Koubun Wakashima.

**Writing – original draft:** Koubun Wakashima, Keigo Asai, Daisuke Kobayashi.

**Writing – review & editing:** Koubun Wakashima, Keigo Asai.

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
