## [Decision Letter · Decision Letter 0]

24 Aug 2020

PONE-D-20-15563

Variables related to fear of novel coronavirus infection (COVID-19) and coping behavior

PLOS ONE

Dear Dr. Wakashima,

Thank you for submitting your manuscript to PLOS ONE. After careful consideration, we feel that it has merit but does not fully meet PLOS ONE’s publication criteria as it currently stands. Therefore, we invite you to submit a revised version of the manuscript that addresses the points raised during the review process.

We look forward to receiving your revised manuscript.

Kind regards,

Ali Montazeri

Academic Editor

PLOS ONE

Journal Requirements:

2. Since one of the key outcomes of your study was the development of a translated version of the FCV-19S, you may wish to include this in the title of your manuscript.

3. PLOS ONE has specific requirements for studies that are presenting a new method or tool as the primary focus, including a newly developed or modified questionnaire or scale (https://journals.plos.org/plosone/s/submission-guidelines#loc-methods-software-databases-and-tools.) One requirement is that the questionnaire or scale must be openly available under a license no more restrictive than CC BY. In light of this, before we proceed, please include a copy of your questionnaire or scale as a Supporting Information file (in the original language) or provide a link if it is available through an online repository.

Reviewers' comments:

Reviewer's Responses to Questions

**Comments to the Author**

1. Is the manuscript technically sound, and do the data support the conclusions?

Reviewer #1: Yes

Reviewer #2: No

2. Has the statistical analysis been performed appropriately and rigorously? 

Reviewer #1: Yes

Reviewer #2: N/A

3. Have the authors made all data underlying the findings in their manuscript fully available?

Reviewer #1: Yes

Reviewer #2: Yes

4. Is the manuscript presented in an intelligible fashion and written in standard English?

Reviewer #1: Yes

Reviewer #2: No

5. Review Comments to the Author

Reviewer #1: This is an interesting and timely study. I have some concerns that will help to improve the manuscript.

1- I would add Japanese or japan in the title

2- Abstract: sampling procedure is missing. Real results and statistics are not reported the abstract. I cannot understand what you really mean on “were relatively comparable to those of the original FCV-19S”? how did you assess coping behavior? Please clearly report mediation results rather than writing a vague interpretation.

3- Introduction is good but needs to improving by adding some information of the latest statistics on covid-19 in japan. Please also add the following references:

Pakpour, A. H., Griffiths, M. D., Chang, K. C., Chen, Y. P., Kuo, Y. J., & Lin, C. Y. (2020). Assessing the fear of COVID-19 among different populations: A response to Ransing et al.(2020). Brain, Behavior, and Immunity.

Pakpour, A. H., & Griffiths, M. D. (2020). The fear of COVID-19 and its role in preventive behaviors. Journal of Concurrent Disorders.

Lin, C. Y. (2020). Social reaction toward the 2019 novel coronavirus (COVID-19). Social Health and Behavior, 3(1), 1.

4- I think that the authors need to compare their results with previous validation version. Please see the following:

Nguyen, H.T.; Do, B.N.; Pham, K.M.; Kim, G.B.; Dam, H.T.; Nguyen, T.T.; Nguyen, T.T.; Nguyen, Y.H.; Sørensen, K.; Pleasant, A.; Duong, T.V. Fear of COVID-19 Scale—Associations of Its Scores with Health Literacy and Health-Related Behaviors among Medical Students. Int. J. Environ. Res. Public Health 2020, 17, 4164.

Soraci, P., Ferrari, A., Abbiati, F. A., Del Fante, E., De Pace, R., Urso, A., & Griffiths, M. D. (2020). Validation and psychometric evaluation of the Italian version of the Fear of COVID-19 Scale. International Journal of Mental Health and Addiction, 1-10.

Haktanir, A., Seki, T., & Dilmaç, B. (2020). Adaptation and evaluation of Turkish version of the fear of COVID-19 scale. Death Studies, 1-9.

Harper, C. A., Satchell, L. P., Fido, D., & Latzman, R. D. (2020). Functional fear predicts public health compliance in the COVID-19 pandemic. International journal of mental health and addiction.

Reznik, A., Gritsenko, V., Konstantinov, V., Khamenka, N., & Isralowitz, R. (2020). COVID-19 fear in Eastern Europe: Validation of the Fear of COVID-19 Scale. International journal of mental health and addiction, 1.

Alyami, Mohsen, Marcus Henning, Christian U. Krägeloh, and Hussain Alyami. "Psychometric evaluation of the Arabic version of the Fear of COVID-19 Scale." International journal of mental health and addiction (2020): 1.

Bitan, D. T., Grossman-Giron, A., Bloch, Y., Mayer, Y., Shiffman, N., & Mendlovic, S. (2020). Fear of COVID-19 scale: Psychometric characteristics, reliability and validity in the Israeli population. Psychiatry Research, 113100.

Winter, Taylor, Benjamin Riordan, Amir Pakpour, Mark Griffiths, Andre Mason, John Poulgrain, and Scarf Damian. "Evaluation of the English version of the Fear of COVID-19 Scale and its relationship with behavior change and political beliefs." (2020).

Sakib, N., Bhuiyan, A. I., Hossain, S., Al Mamun, F., Hosen, I., Abdullah, A. H., ... & Sikder, M. T. (2020). Psychometric validation of the Bangla Fear of COVID-19 Scale: Confirmatory factor analysis and Rasch analysis. International Journal of Mental Health and Addiction.

Reviewer #2: The introduction is not convincing. It fails to show the current literature gap. The aim of the study is not clear. The study should clearly specify the mentioned issues; whether you are validating FCV-19S or developing a model of COVID-19 fear effects…?

Method: how did you invite the participants to the study? How many were approached? What were the main reasons for non-participation? The rational for study sample size should be stated. The rationale behind selecting the variables included in the questionnaire should be stated.

The statistical analysis should be clearly explained. The conceptual model that guided the authors for analysis should be explained in details.

Table 1. please indicate which test was used for analysis.

Table 2: writing error, heart races!

Why did you select model 4 as the final model? Did you change any item in the scale? What modification/s was/were made in to the scale?

Why did you use HAD for structural validity among a community population?

What do you mean by (α= .93/ ω= .93) in table 3? These should be explained.

The main concern is that, could we use the same sample for both validation study and SEM?

What are the differences between table 4 and 5?

The discussion is highly poor. In this part you should compare the results obtained from your study with similar studies either confirming or rejecting the current results. Then, express your explanations or justifications.

6. PLOS authors have the option to publish the peer review history of their article (what does this mean?). If published, this will include your full peer review and any attached files.

Reviewer #1: **Yes: **Amir Pakpour

Reviewer #2: **Yes: **Marzieh Aaraban

---

## [Author Response · Author response to Decision Letter 0]

1 Oct 2020

Reviewer #1 :

Comment 1 : Title

I would add Japanese or japan in the title.

Response:

We agree with your opinion. We have changed the title from “Variables related to fear of novel coronavirus infection (COVID-19) and coping behavior” to “The Japanese version of the Fear of COVID-19 scale: Reliability, validity, and relation to coping behavior”.

Comment 2 : Abstract

Sampling procedure is missing. Real results and statistics are not reported the abstract. I cannot understand what you really mean on “were relatively comparable to those of the original FCV-19S”? how did you assess coping behavior? Please clearly report mediation results rather than writing a vague interpretation.

Response:

We agree with the reviewer’s point. We have made the following modifications.

(1) We have added “450 Japanese participants were recruited from a crowdsourcing platform” to the Abstract as the sampling procedure (p.2 lines 21-22).

(2) We have added real results and statistics to the Abstract (p.2 lines 28–33).

(3) We have changed the sentence from “were relatively comparable to those of the original FCV-19S” to “These results suggest that the Japanese FCV-19S is a psychometric scale with the same reliability and validity as the original FCV-19S.”

Comment 3 : Introduction

Introduction is good but needs to improving by adding some information of the latest statistics on covid-19 in japan. Please also add the following references:

Response:

To page 3 of the revised manuscript, we have included the latest statistics on COVID-19 from the entire world and from Japan. We have added references to lines 53–54 (Lin, 2020), lines 56–57 (Pakpour and Griffiths, 2020), and lines 62–63 (Pakpour et al., 2020).

Comment 4 :

I think that the authors need to compare their results with previous validation version.

Response:

Thank you for introducing us to the many references. We have cited nine references to support the validation lines 60–73. These are the following: Sakib et al. (2020), Bitan et al. (2020), Soraci et al. (2020), Winter et al. (2020), Reznik et al. (2020), Alyami et al. (2020), Haktanir et al. (2020), Nguyen et al. (2020), and Harper et al. (2020).

 

Reviewer #2 :

Comment 1 : Introduction

The introduction is not convincing. It fails to show the current literature gap. The aim of the study is not clear. The study should clearly specify the mentioned issues; whether you are validating FCV-19S or developing a model of COVID-19 fear effects…?

Response:

The reviewer has commented on the lack of persuasiveness of the Introduction and the lack of clarity of purpose. We agree with the reviewer’s points. We have particularly cited the references introduced by Reviewer 1 (lines 60–73) and have described the purpose as follows: The purpose of this study was twofold. First, this study translates FCV-19S established by Ahorsu et al. [10] into Japanese and assesses reliability and validity in Japan based on a procedure equivalent to that used by Ahorsu et al. [10]. Secondly, this study develops a model of the effects of COVID-19 fear on coping behaviors, including the reasons for the behavior (p. 5 lines 82–85).

Comment 2 : Method

How did you invite the participants to the study? How many were approached? What were the main reasons for non-participation? The rational for study sample size should be stated.

Response:

The reviewer has commented on the lack of clarity related to data collection in the Materials and Methods section. We have specifically added information related to how we recruit participants and the reasonableness of our sample size (Page 5, lines 89–101). Because we used crowdsourcing and decided to end the study when the number of participants reached 450, we cannot state the reasons for non-participants.

Comment 3 :

The rationale behind selecting the variables included in the questionnaire should be stated.

Response:

We agree with the reviewer’s point. We have stated the rationale for our choice of variables by citing earlier studies (Page 6, lines 118–120; Page 7, lines 132–134).

Comment 4 :

The statistical analysis should be clearly explained. The conceptual model that guided the authors for analysis should be explained in details.

Response:

Thank you for your comment. We realized that our original explanation was unclear and revised as follows:

(1) The Data analysis section described all the analyses used for this study (lines 201–230).

(2) Regarding the conceptual model, we discussed the need to measure the reasons for behavior in the Introduction section and the relation between FCV-19S, coping behavior, and reasons for behavior, as assumed in this study (lines 74–81). In addition, the conceptual model was explained in the Data analysis section (lines 222–230).

Comment 5 :

Table 1. please indicate which test was used for analysis.

Response:

We have provided a brief description of the analysis of the Table 1 footnote of revised manuscript. Note: ** p<0.01, sex, smoking habit, diseases being treated, living with family and presence of persons infected were subjected to t-tests, and analysis of variance was performed for age, health condition, work status, and sources of information.

Comment 6 :

Table 2: writing error, heart races!

Response:

Thank you for pointing out this error. An earlier manuscript did not present all sentences in Item 7. We have therefore added all sentences to item 7. The original FCV-19S (Ahorsu et al., 2020) also uses the word "heart races".

Comment 7 :

Why did you select model 4 as the final model? Did you change any item in the scale? What modification/s was/were made in to the scale?

Response:

The reviewer has commented on the lack of clarity related to the final model decision. We decided to use Model 4 as the final model because it was an acceptable fit over the other two improved models (Model 2 and Model 3) (lines 242–243). We have added a description of MI to the Data analysis section to show that we have assumed error correlation related to MI (lines 213–215). Some other studies of FCV-19S have also assumed error correlations (Alyami et al., 2020; Haktanir et al., 2020) (lines 335–337). We did not change the items in the scale.

Comment 8 :

Why did you use HAD for structural validity among a community population?

Response:

HAD has been used in the validation of FCV-19S and has been found to be reliable and valid in studies of community samples. We have added the following explanation to lines 166–168 of the revised manuscript. Although the HADS [35] was designed originally for patients in nonpsychiatric hospital clinics, it has been found to be reliable and valid in general samples [34]. HADS was positively correlated with FCV-19S [10].

Comment 9 :

What do you mean by (α= .93/ ω= .93) in table 3? These should be explained.

Response:

We have provided a brief description of � and � in the Table 2 and Table 3 footnotes. We have also given the use of � and � as indicators of reliability in Data analysis section (lines 215–216).

Comment 10 :

The main concern is that, could we use the same sample for both validation study and SEM?

Response:

The reviewer has asked about the propriety of using the same sample for validation and SEM.

Many other studies have applied factor analysis and other analyses (including SEM) to the same samples. Therefore we believe there is no particular problem with their application here.

Comment 11 :

What are the differences between table 4 and 5?

Response:

Table 4 presents results of exploratory factor analysis for “coping behavior.” Table 5 shows the results of exploratory factor analysis for “reason for behavior.” For “reason for behavior,” we changed the factor names to clarify it that it is a reason (Previous manuscript: self-determination and conformity behavior. Revised manuscript: self-determining reason and conformity reason).

Comment 12 :

The discussion is highly poor. In this part, you should compare the results obtained from your study with similar studies either confirming or rejecting the current results. Then, express your explanations or justifications.

Response:

The reviewer is concerned about the lack of sufficient discussion. The reviewer is correct. We appreciate the chance to clarify our exposition. We have revised the paper as explained below.

(1) Results of the confirmatory factor analysis are discussed in comparison to results obtained for other countries (lines 335–346).

(2) The relation between sociodemographic variables and FCV-19S was discussed in comparison to the results of earlier studies (lines 349–358).

(3) Effects of FCV-19S on coping behaviors were discussed in comparison to Winter et al., 2020 and to Harper et al., 2020 (lines 359–375 ).

 

Other points to change 

ERROR CORRECTED

p.6 line 103: “Socio-demographic variable” (previous manuscript: Attributes of the respondents)

p.6 line 108: “6 = having other symptoms” (previous manuscript: 9 = having other symptoms)

p.11 line 236: “FCV-19S-J” (previous manuscript: Japanese FCV-19S-J)

p.23 line 389: “longitudinal study” (previous manuscript: cross-sectional surveys)

---

## [Decision Letter · Decision Letter 1]

26 Oct 2020

The Japanese version of the Fear of COVID-19 scale: Reliability, validity, and relation to coping behavior

PONE-D-20-15563R1

Dear Dr. Wakashima,

We’re pleased to inform you that your manuscript has been judged scientifically suitable for publication and will be formally accepted for publication once it meets all outstanding technical requirements.

Kind regards,

Ali Montazeri

Academic Editor

PLOS ONE

Additional Editor Comments (optional):

Reviewers' comments:

Reviewer's Responses to Questions

**Comments to the Author**

1. If the authors have adequately addressed your comments raised in a previous round of review and you feel that this manuscript is now acceptable for publication, you may indicate that here to bypass the “Comments to the Author” section, enter your conflict of interest statement in the “Confidential to Editor” section, and submit your "Accept" recommendation.

Reviewer #1: All comments have been addressed

2. Is the manuscript technically sound, and do the data support the conclusions?

Reviewer #1: Yes

3. Has the statistical analysis been performed appropriately and rigorously? 

Reviewer #1: Yes

4. Have the authors made all data underlying the findings in their manuscript fully available?

Reviewer #1: Yes

5. Is the manuscript presented in an intelligible fashion and written in standard English?

Reviewer #1: Yes

6. Review Comments to the Author

Reviewer #1: The authors have addressed my comments. The paper can be published. Thank you so much for sending the paper for review in PLOS One.

7. PLOS authors have the option to publish the peer review history of their article (what does this mean?). If published, this will include your full peer review and any attached files.

Reviewer #1: No

---

## [Editor Report · Acceptance letter]

28 Oct 2020

PONE-D-20-15563R1 

The Japanese version of the Fear of COVID-19 scale: Reliability, validity, and relation to coping behavior 

Dear Dr. Wakashima:

I'm pleased to inform you that your manuscript has been deemed suitable for publication in PLOS ONE. Congratulations! Your manuscript is now with our production department. 

Kind regards, 

on behalf of

Professor Ali Montazeri 

Academic Editor

PLOS ONE